# Reading Recognition in the Wild

**Charig Yang[1,2], Samiul Alam[3], Shakhrul Iman Siam[3], Michael J. Proulx[1], Lambert Mathias[1],
Kiran Somasundaram[1], Luis Pesqueira[1], James Fort[1], Sheroze Sheriffdeen[1], Omkar Parkhi[1],
Carl Ren[1], Mi Zhang[3], Yuning Chai[1], Richard Newcombe[1], Hyo Jin Kim[1]**

[1]Meta Reality Labs Research    [2]VGG, University of Oxford    [3]The Ohio State University
charig@robots.ox.ac.uk, kimhyojin@meta.com

https://www.projectaria.com/datasets/reading-in-the-wild/

## Abstract

To enable egocentric contextual AI in always-on smart glasses, it is crucial to be able
to keep a record of the user's interactions with the world, including during reading.
In this paper, we introduce a new task of *reading recognition* to determine *when*
the user is reading. We first introduce the first-of-its-kind large-scale multimodal
*Reading in the Wild* dataset, containing 100 hours of reading and non-reading videos
in diverse and realistic scenarios. We then identify three modalities (egocentric
RGB, eye gaze, head pose) that can be used to solve the task, and present a flexible
transformer model that performs the task using these modalities, either individually
or combined. We show that these modalities are relevant and complementary to
the task, and investigate how to efficiently and effectively encode each modality.
Additionally, we show the usefulness of this dataset towards classifying types of
reading, extending current reading understanding studies conducted in constrained
settings to larger scale, diversity and realism. Code, model, and data will be public.

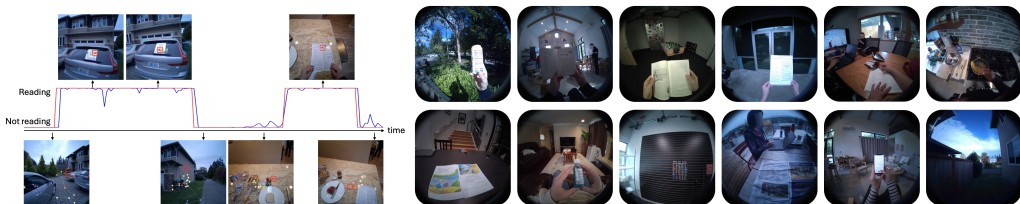

Figure 1: **Am I reading?** The left figure shows a timeline as the user navigates the world. We aim to solve the
task of reading recognition to enable AI assistants in always-on wearables. We identify three modalities: eye
gaze (in colored dot patterns), RGB crop around gaze (in red box), and inertial sensors performs the task to high
accuracy (with Prediction and GT shown). Images from our *Reading in the Wild* dataset, which features 100
hours of diverse reading and non-reading activities in real-world settings, with examples shown in the right.

## 1 Introduction

The potential future of AI personal assistants depends on its ability to understand the physical context
of the user. Smart glasses are becoming a promising device form factor capable of linking visual AI
capabilities to the real world. Recently, there has been a sharp rise in the development of smart glasses,
both products (Meta Ray-Ban, Amazon Echo Frames) and prototypes (Snapchat Spectacles, Halliday
AI Glasses, Xreal One Pro). These all-day wearable devices enable proactive, personalized, and
contextualized AI agents to perceive the world like humans do by understanding the users' context.

However, for always-on wearable glasses, due to both *hardware* (power, bandwidth, heat) and *software*
(perception capability of AI agents, especially with heavy models) constraints, it is impractical to

39th Conference on Neural Information Processing Systems (NeurIPS 2025).

record and process every single frame over long periods of time. One solution is to have a proxy signal, so that the device can record and process *key* frames only *when* relevant. The question becomes: what forms important context of the user that the AI assistant needs to know, and how do we know when to capture them?

The ability to read underpins one of, if not the most important unique modalities by which modern humans communicate, entertain each other, and learn. Reading is a key mechanism humans use to communicate with high fidelity and high information density. Reading spans a broad array of mediums, from handwritten and printed text on paper and digital displays to environmental signposts. The act of reading occurs within real-time communication with one another and today's AI chatbots, through to reading long-form articles in books or online. Enabling AI with the ability to recognize reading is hence clearly one of the most important context signals a future AI can be enabled with to unlock truly personalized and contextually relevant AI.

Given this, we ask: how can we provide future AI with the ability to know when someone is reading? This apparently simple idea underpins the ability to efficiently enable devices to know what the user has and has not read, and hence where they can assist given what it understands that the user has read.

This task of *reading recognition* is challenging for two main reasons. First, the problem can often be ill-posed: just because a text exists in the field of view does not mean that the user is reading it (or even looking at it), which is ambiguous to solve using visual information alone. Also, the method should be efficient for real-time, always-on computation subject to the practical constraints of a wearable device. Both of these challenges render OCR-based text detection methods impractical, given the inability to solve the ambiguity and the requirements for high-resolution capture and processing. Instead, reading recognition can be used as an efficient proxy to indicate *when* and *where* it is relevant to invoke heavier models (OCR and VLMs) instead of running these models all the time.

Motivated by this question, we introduce a new dataset created with Project Aria [11] glasses, which enables us to develop the contextual AI capability of detecting *when* a wearer is reading. We present the first-of-its-kind large-scale multimodal "Reading in the Wild" dataset, containing 100 hours of reading and non-reading videos in diverse and realistic scenarios. This dataset allows us to identify three modalities (egocentric RGB, eye gaze, head pose) that can be used to solve the task. We then present a flexible transformer model that performs the task using these modalities, either individually or combined. We show that these modalities are relevant and complementary to the task and investigate how to efficiently and effectively encode each modality, as well as the model's ability to generalize towards unseen scenarios and perform real-time reading detection.

Achieving reading recognition makes it feasible to keep a record of a user's reading interactions with the world to build a contextually aware AI. It also enables several other applications: it allows reading assistant tools [38] in children with learning difficulties [5] and people with low vision [40] to operate in the real world; it can also be used to track whether a user has read crucial information (*e.g.* signs during driving) and to measure attention and distraction while performing a task.

Additionally, the dataset and method contributed in this paper can be extended to classifying different types of reading. This has been of interest in cognitive studies in reading comprehension, but they are often limited to controlled environments [25, 26, 21, 1, 7], hence limiting its usefulness. We show that our dataset allows for reading mode and medium classification to be performed in unconstrained settings, and provide experimental results in this direction.

In summary, we make the following contributions:

- First, we introduce a new task of reading recognition *in the wild*, and demonstrate its usefulness. Unlike previous studies, we focus on in-the-wild settings and practicality towards wearable glasses.
- Second, we present the first-of-its-kind large-scale egocentric multimodal *Reading in the Wild* dataset, which will be made publicly available, alongside a scalable protocol for data collection.
- Third, we identify three modalities relevant and complementary to the task (RGB, gaze, and IMU), and develop a lightweight, flexible model that inputs these modalities either individually or in combination for reading recognition, resulting in a strong and efficient baseline for this task.
- Fourth, we show that our method and dataset extend towards reading understanding, including classifying reading mode and medium, demonstrating usefulness towards cognitive studies.

| Subset | Size | Indoor | Outdoor | Medium | Text type | Multi-task | Mode | Language | Not reading | Mixed |
|---|---|---|---|---|---|---|---|---|---|---|
| **Seattle** (train/val/test) **Focus**: diversity | 80 hours 81 people 1061 videos | Offices Libraries Homes Stores | Balconies Patios Roads/trails In the woods | Print Digital Objects | Paragraphs Short texts Non-texts Dynamic texts | None Walking Writing Typing | Engaged Skimming Scanning Out loud | English ($\rightarrow$) | Daily activities Hard negatives (71%/29%) | Alternating sequences (reading / not reading) |
| **Columbus** (test) **Focus**: edge cases, generalization | 20 hours 31 people 655 videos | Offices Libraries Lounges Corridors | | Print Digital Objects | Paragraphs Short texts Non-texts | None | Engaged Scanning | English ($\rightarrow$) Bengali ($\rightarrow$) Chinese ($\downarrow$) Arabic ($\leftarrow$) | Hard negatives Daily activities (58%/42%) | Mirror setups (same settings, one reading, another not) |

Table 1: **Dataset overview.** We separately collect two subsets for the dataset. Seattle subset focuses on diversity, while Columbus subset looks at the model's generalization towards unseen settings, as well as edge cases where the model fails. See Appendix A for more details.

## 2 Related Work

**Reading recognition** has been a long studied task with rich literature. Eye gaze has been used as the primary signal [21, 6, 1, 26], however, it relied on handcrafted feature engineering methods such as detecting fixations and saccades, which we show are unnecessary. Moreover, the experiments are usually constrained, and not performed *in the wild*. Other modalities have also been considered, such as electrooculography (EOG) signals [4], though the usage of electrodes can be invasive and hence less practical towards building user-friendly wearable glasses. In this paper, we steer this towards practical usage in modern smart glasses, where we show that gaze can be used in combination with visual information and IMU sensors. With recent advances in wearable devices, reading recognition expands to tasks such as word recognition and reading order prediction [17]. While this is relevant, it concerns the reading content, and assumes the user is already reading, which differs from the task of detecting whether the user is reading in this paper. Applications include reading comprehension [31, 19, 9, 36], understanding user behavior [7], and in building reading assistants [38, 5, 40]. However, the literature is largely constrained to controlled environments.

**Egocentric activity recognition** is a popular vision task that usually require computationally heavy solutions using video input [20, 45]. In terms of data, reading is only a subset of activities in some common datasets such as EGTEA Gaze+ [27] and Ego-Exo4D [14]. However, not all datasets contain reading [10]. For those which include reading, its nature is very restricted to activities such as reading recipes (in [27]), reading covid testing manuals, climbing instructions, and music sheets (in [14]). Ego4D [13] offers a more diverse range of reading activities, but only less than 1% of the data includes eye gaze. In contrast, our paper focuses on efficient reading recognition, and the proposed dataset contains large-scale and diverse reading and non-reading examples with eye gaze information.

**Gaze in computer vision** has started to gain popularity, where gaze has many applicable uses. One popular route is to perform gaze prediction *i.e.* predicting where the user is looking at [34, 12, 32, 37] or how the user interacts with objects he/she observes [18, 39, 29]. In medical applications, eye gaze can be used as a saliency test to ensure integrity in medial image analyses [41, 23, 22, 30], as well as predicting learning disorders [16]. Recently, gaze has also been used to complement vision, such as in action recognition [44], narration [8], and vision-language models [24]. Our paper further explores whether gaze can *reduce* the input requirements for computer vision models by only using gaze and/or parts of vision that are associated with gaze instead of using the whole image sequence.

## 3 Reading in the Wild Dataset

### 3.1 Overview

The dataset contains about 100 hours of recordings of reading and non-reading activities collected from one RGB (30Hz, 1408p, 110° FoV) and two SLAM (150° FoV) cameras, two eye tracking cameras (60Hz, calibrated), two IMUs (with odometry outputs from visual SLAM), and audio transcribed using WhisperX [2]. We independently collect two subsets of this dataset, as in Table 1.

**Seattle** is collected for training, validation, and testing. We mainly focus on collecting reading and non-reading activities in diverse scenarios, in terms of participants' identities, reading scenarios, reading modes, and reading materials. It contains a mix of normal and hard examples, as well as mixed sequences alternating between reading and non-reading activities. The dataset is collected in homes, office spaces, libraries, and outdoors.

**Columbus** is collected to find out where the model breaks in zero-shot experiments. It contains examples of hard negatives (where text is present but is not being read), searching/browsing (which gives confusing gaze patterns), and reading non-English texts (where reading direction differs).

| Dataset | Gaze | RGB | Reading | Real | HN |
|---|---|---|---|---|---|
| Ego4D | ✗ | ✓ | Limited | ✓ | ✗ |
| Ego-Exo4D | 10Hz | ✓ | Limited | ✓ | ✗ |
| EGTEA | 30Hz | ✓ | Limited | ✓ | ✗ |
| ZuCo | 500Hz | ✗ | ✓ | ✗ | ✗ |
| InteRead | 1.2kHz | ✗ | ✓ | ✗ | ✗ |
| Ours | 60Hz | ✓ | ✓ | ✓ | ✓ |

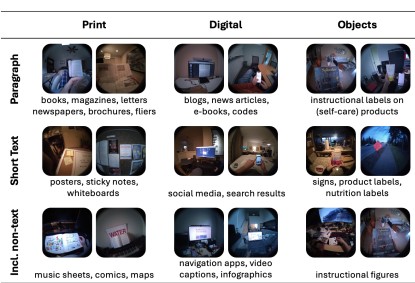

Table 2: **Comparison to existing datasets**. Our dataset is the first reading dataset that contains high-frequency eye-gaze, diverse and realistic egocentric videos, and hard negative (HN) samples.

Figure 2: **Diversity in reading materials.** Reading examples across different materials, both text type (rows) and medium (column).

## 3.2 Comparison to existing datasets

The closest kins to our dataset come in two categories, as shown in Table 2. First, in egocentric video datasets [14, 13, 27], there are very limited reading sequences and they lack diversity as each dataset only reads from 1-2 examples (COVID test kits for Ego-Exo4D, recipes for EGTEA). Moreover, their eye tracking frequencies are also limited. Second, there are cognitive studies that focus on human gaze behavior during reading [15, 43] with high-frequency eye tracking. However, these studies are conducted in very constrained scenarios such as reading a text in front of a screen. Moreover, these studies only collect gaze data without RGB stream.

## 3.3 Contents

**Reading.** Our dataset presents a large diversity in reading activities, including:

- **Reading mode**: Our dataset contains different reading modes, including deep reading (careful, engaged reading), skimming (quickly glancing through for general ideas), scanning (searching for specific information), and reading aloud (verbalizing the text).
- **Single/Multi-task reading**: Our dataset not only covers single-task reading, where the focus is solely on the reading material, but also reading while multitasking, such as reading while writing, typing, or walking.
- **Medium and text type**: We collect data across mediums: print (books, newspapers, flyers), digital (phones, monitors), everyday objects (product labels, whiteboards); and text types: paragraphs, short texts, non-texts, and dynamic texts (video captions and subtitles) as illustrated in Figure 2.
- **Demographics**: We collect data among 111 participants and include their age range and gender.
- **Location**: For diversity, we collect scenes across indoor (e.g., meeting rooms, bedrooms, living rooms), balconies, outdoors, and in the woods.

**Non-reading.** We also collect negative examples. This includes *Everyday activities* that do not involve reading such as physical exercise, outdoor activities, creative arts, culinary activities, and household chores, as well as *Hard negatives*, where text is present in the scene but is not being read, which would confuse RGB-only models.

**Mixed**. We also collect *Alternating sequences*, where the participants alternating between reading and non-reading with annotated timestamps, and *Mirror setups* where we have the same participant perform reading and non-reading activity in the same environment and the same material.

## 3.4 Data collection process

**Logistics.** We recruited a total of 111 participants, targeting a uniform distribution for gender and age. We gave each participant a list of tasks to record, with moderators monitoring to ensure that the recordings are correct as desired.

**Instructions.** We divided the collections into tasks, each with specific instructions, as elaborated in the Appendix. We also asked the participants to perform eye gaze calibration within each recording.

**Privacy.** We strictly followed Project Aria Research guidelines. All data has been de-identified, and faces and license plates were anonymized with EgoBlur [35]. We source the venues ourselves do not use the participants' private spaces to prevent exposure of sensitive or identifiable information.

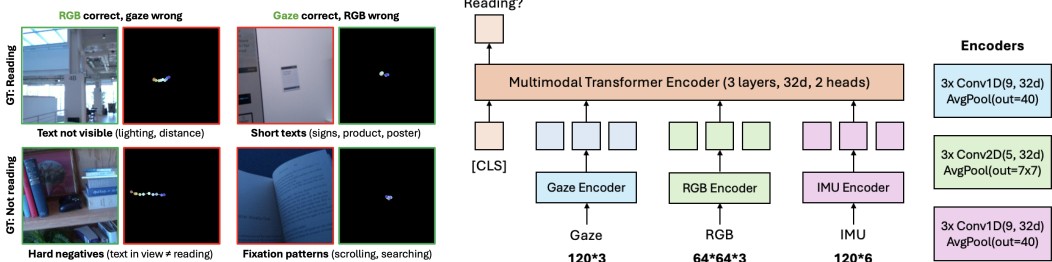

Figure 3: **Complementary modalities**. Example success and failure cases for gaze and RGB, suggesting the benefit of multimodality.

Figure 4: **Model architecture**. Our model is a simple transformer encoder with any combination of gaze, RGB, and IMU as input.

**Scalable Protocol through Automatic labeling.** In addition to the dataset itself, we also present a protocol for scalable, high-quality data collection. Instead of manually labeling the timestamps, we instruct the participants to say "start reading!" whenever they start reading, and "finished reading!" whenever they finish. In doing so, we can simply use WhisperX [2] to obtain accurate timestamps without requiring manual annotations.

**Quality assurance.** We have several protocols to ensure that participants have read the text. This involves before (pre-reading questions) and after (post-reading questions and summarization). For the subset where the user reads out loud, the audio transcription can also be used to for quality assurance.

## 4 Method

### 4.1 Task definition

Formally, at time $t$, we want to predict the confidence score $s_t \in [0, 1]$ whether the user is reading or not, given several input modalities: eye gaze patterns $g_{t-T \leq \tau \leq t} \in \mathbb{R}^{f \times T \times d}$, instantaneous RGB $I_t \in \mathbb{R}^{H \times W \times C}$ and head pose (IMU) sensor readings $z_{t-T \leq \tau \leq t} \in \mathbb{R}^{f \times T \times d}$, where $f$ is the sampling frequency and $T$ is the input duration *i.e.* $s_t = \Phi(g_{t-T \leq \tau \leq t}, I_t, z_{t-T \leq \tau \leq t})$. Each modality has different advantages and drawbacks. To harness the strength of all modalities, we propose a multimodal model that takes into account all three modalities as input. In the following sections, we first discuss individual modalities, followed by the model architecture.

### 4.2 Input modalities

**Gaze.** There exists a vast literature suggesting that gaze can be used to detect reading activity without visual information [21, 6, 1, 26]. However, their experiments are limited to constrained environments (reading long paragraphs in front of a screen), and they rely on feature engineering methods such as fixation detection to circumvent small-scale data. As we demonstrate in the experiments section, training on diverse data translates well to open-world settings, and feature engineering is unnecessary at scale, which makes it robust to low frequency eye tracking inputs.

**RGB.** As with action recognition methods, visual information has been an effective cue in the computer vision community. However, processing video models on a wearable device is expensive. Meanwhile, there has been an interest in using gaze to guide model attention in action recognition [27, 14, 44]. For reading, we argue that region outside the gaze point is likely to be irrelevant, as the high-resolution human fovea capable of reading only covers a small region (2°) around the gaze [33]. Therefore, we only crop the image around the gaze region. This also allows for large efficiency gains as capture and processing only needs to be done on a small patch. We find that cropping using only 1/484 of an image (64px, 5° from 110° FoV) can result in good accuracy, with the remainder for context and gaze uncertainties.

**Head pose (IMU).** We also explore using odometry measurements. While not a good indicator on its own, we find that it helps as a secondary sensor. The intuition here is that some inertial motions can be used to address ambiguities, such as distinguishing between reading and horizontal head motion.

**Complementary modalities.** The main reason for using multiple modalities is that they are complementary: they excel and fail in different places. For example, eye gaze can perform well even if the text is not visible due to lighting or distance that images sometimes miss out, while RGB works in cases where gaze patterns are not obvious, such as when reading short texts like signs, as

shown in Figure 3. While IMU is not strong on its own, we show later that it further provides cues to disambiguate some cases (e.g. turning heads vs reading).

## 4.3 Model

In order for this to be practical towards always-on wearable devices, we propose a simple and efficient model that achieves a strong practical baseline for the task. Particularly, we propose a flexible multimodal transformer model that takes in different modalities as input, as shown in Figure 4. By keeping the model simple, we can investigate different combinations and forms of modalities.

**Input.** Unless otherwise stated (such as in ablation studies), we use $T = 2$, $f = 60$ for 3D eye gaze and 6DoF IMU, and a 5° FoV ($H, W = 64$) crop for RGB as default.

**Modality encoder.** The model consists of different encoders $\Phi_{\{g,r,i\}}$ (where g,r,i represent gaze, RGB, and IMU respectively) to tokenize individual modality into feature tokens $f_{\{g,r,i\}} \in \mathbb{R}^{N \times D}$. We use three layers each of 1D (gaze and IMU) and 2D (RGB) convolutions.

**Multimodal transformer.** We then combine these feature tokens using a simple transformer encoder $\Phi_t$ and a linear head over the [CLS] token *i.e.* $s_t = \Phi_t(f_g, f_r, f_i)$.

**Modality dropout.** During training, we dropout entire modalities at random, which serves two purposes: (i) it helps with training less-used modalities; (ii) during inference, the model can perform well even without all modalities being present.

## 4.4 Generalization

While we train on English texts, we find that our model generalizes well to other left-to-right languages across different writing systems, but struggles with vertical and right-to-left texts, as the gaze pattern is in a different direction. To address this, we find that simply augmenting the gaze at inference time (90° rotation for vertical texts and horizontal flip for right-to-left texts) allows the model to generalize well. In practical scenarios, this can be done depending on geo-location. During training, we also add a small fraction of rotated gaze to help with reading vertical texts.

# 5 Experimental Setup

## 5.1 Dataset split

We split the Seattle subset into training, validation, and test sets, and train the model on the training set. We evaluate on (i) the test set of the Seattle subset, and (ii) the entire Columbus subset. We also evaluate on specific subsets to study latency and generalization.

## 5.2 Implementation details

**Model.** For the encoders, we use three layers of 1D convolution (kernel size 9, 32 dims) for gaze and IMU, and three layers of 2D convolution (kernel size 5, 32 dims) for RGB. We then feed the tokens as input to three layers of transformer encoder (32 dims, 2 heads) before linearly projecting the [CLS] token to two classes. The combined model is lightweight, with 137k parameters.

**Training.** We impose modality dropout such that there is an equal probability of using one, two, or three modalities at the same time, as well as perform rotation augmentation. We use Adam optimizer with learning rate $1e^{-3}$ for ten epochs. All models are trained using a single GPU. The code and models will be released alongside the dataset.

## 5.3 Evaluation metrics

**Classification metrics.** We calculate the accuracy and F1 scores for each task at 0.5 confidence threshold. We also vary this threshold, and report the precision at 0.9 recall (denoted as $P_{R=.9}$).

**Latency.** We consider latency to be the time between a state change and model detecting it, and is unrelated to the computational time, which we assume to be negligible given the small model size.

# 6 Results

## 6.1 Main results

We present the main results and visualizations in Figure 5.

| Gaze | RGB | IMU | Acc | F1 | $P_{R=.9}$ |
|:---:|:---:|:---:|:---:|:---:|:---:|
| ✓ | | | 82.3 | 84.5 | 79.8 |
| | ✓ | | 82.2 | 83.7 | 76.5 |
| | | ✓ | 74.7 | 80.0 | 71.9 |
| ✓ | | ✓ | 84.9 | 86.5 | 83.6 |
| | ✓ | ✓ | 83.5 | 85.2 | 82.3 |
| ✓ | ✓ | | 86.0 | 87.8 | 87.3 |
| ✓ | ✓ | ✓ | **86.9** | **88.1** | **88.0** |

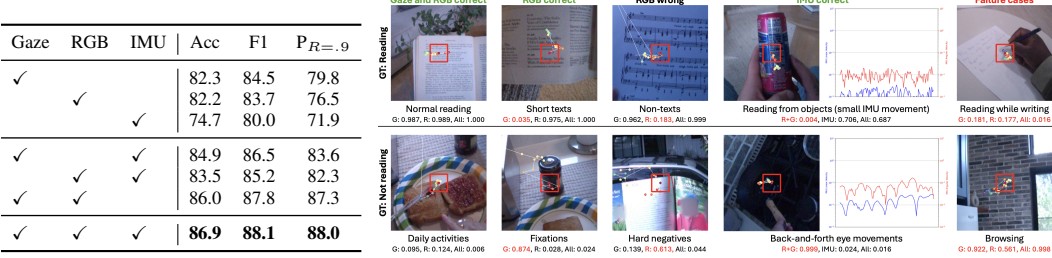

(a) Main results  (b) Visualization (**G/R** are for gaze/RGB, with wrong ones in red)

Figure 5: **Main results and visualizations.** We show the results on Seattle (test set). (a) Our method performs the task to good accuracy, and combining all modalities yields the best results. Metrics are accuracy and F1 score at 0.5 threshold, and precision at 0.9 recall. (b) We show: (i) Col. 1, banal success cases distinguishing reading from daily activities; (ii) Col. 2-4, difficult cases where our combined model predicts correctly even if individual modality fails, including reading from objects, short texts, non-texts, fixation patterns, and hard negatives; (iii) Col. 5, failure cases where all modalities fail, including reading while writing and browsing.

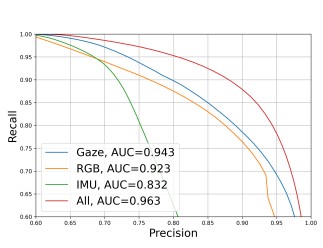

| Scenario | Acc (%) |
|:---|:---:|
| Digital media (normal scenarios) | 95.3 |
| Print media (normal scenarios) | 93.8 |
| **Reading average** | **88.1** |
| Objects (normal scenarios) | 87.6 |
| Reading while walking | 81.4 |
| Reading from videos | 78.0 |
| Reading non-texts | 65.8 |
| Reading while writing/typing | 55.5 |
| Daily activities | 95.2 |
| **Not reading average** | **86.4** |
| Hard negatives | 74.7 |

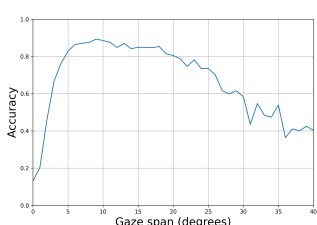

(a) Precision-recall curve  (b) Breakdown by scenario  (c) Breakdown by gaze span

Figure 6: **Results breakdown.** We present the breakdown for the main results, including (a) precision-recall curve for different modalities (b) breakdown by scenario to highlight difficult cases (c) breakdown by gaze span.

**Single modality.** We find that gaze and RGB are able to achieve reasonable performance individually, and their performances are similar to each other (82.3% and 82.2% accuracy respectively). However, as shown in the visualizations, they have different success and failure cases. IMU alone does not perform very well, which is reasonable, as the problem becomes ill-posed, and the model can only guess the lack of motion as not reading (and vice versa).

**Combined modalities.** We find that IMU monotonically improves upon gaze (+2.6%) or RGB (+1.3%) as secondary modality, with small extra compute. Qualitatively, we see that IMU helps improve several corner cases, and RGB is particularly strong for short texts. We also find that all modalities combined yields the best performance of 86.9% in accuracy (+4.6% from best single-modality model), validating the complementary roles of different modalities.

## 6.2 Results breakdown

We show the breakdown of results in Figure 6.

**Scenario breakdown.** We break down the results of the combined model. We find that the model mostly succeeds in normal cases, but fails in cases where reading is atypical, such as reading non-texts (maps, music sheets), or when reading while writing or typing. The model also struggles with hard negative examples introduced in this dataset.

**Gaze span breakdown.** We also break down the results of reading sequences by the horizontal gaze field of view, as it correlates with text size. We find that the accuracy is the highest (86.1%) for fields of view of 5-20°, corresponding to 64-256 pixels, with accuracy dropping sharply for both below (59.3%) and above (70.6%) this range.

## 6.3 Generalization

We use the model trained on the Seattle subset to evaluate on unseen scenarios, shown in Table 3.

**Zero-shot generalization.** To evaluate zero-shot capabilities, we test on the separately collected Columbus subset. We show that the model performs reasonably zero-shot, and draw similar conclu-

Table 3 (a) Zero-shot on Columbus

| Gaze | RGB | IMU | Acc | F1 | $P_{R=.9}$ |
|---|---|---|---|---|---|
| ✓ | | | 77.1 | 84.0 | 84.1 |
| | ✓ | | 76.7 | 84.5 | 83.4 |
| ✓ | ✓ | | 82.8 | 88.7 | **88.2** |
| ✓ | ✓ | ✓ | **82.9** | **88.8** | **88.2** |

Table 3 (b) Cross-language (text direction)

| Language | Aug | Acc | F1 |
|---|---|---|---|
| English → | - | 81.2 | 87.0 |
| Bengali → | - | 93.0 | 95.9 |
| Chinese ↓ | - | 35.5 | 51.6 |
| | rotate | 85.1 (+49.6) | 91.9 (+40.3) |
| Arabic ← | - | 21.0 | 23.8 |
| | flip | 51.5 (+30.5) | 63.8 (+40.0) |

Table 3 (c) Generalization to EGTEA

| Test | Train | Acc | F1 |
|---|---|---|---|
| Seattle | Seattle | 79.3 | 81.2 |
| | EGTEA | 62.9 (-16.4) | 56.9 (-24.3) |
| EGTEA | EGTEA | 89.6 | 70.6 |
| | Seattle | 87.7 (-1.9) | 63.4 (-7.2) |

Table 3: **Generalization results.** Using model trained on Seattle subset, we test on (a) separately collected Columbus subset; (b) different languages with different reading patterns and direction (despite only being trained with English), where we explore using rotation and flipping augmentations; (c) cross-generalization with EGTEA. The model generalizes one way (Seattle → EGTEA) but not the other.

Figure 7 (a) Latency

| Gaze | RGB | IMU | Acc | F1 | Latency (s) |
|---|---|---|---|---|---|
| ✓(1s) | | | 77.1 | 75.6 | 0.526 |
| ✓(2s) | | | 79.0 | 78.9 | 0.831 |
| ✓(3s) | | | 79.3 | 77.8 | 1.013 |
| | ✓ | | 73.8 | 68.7 | **0.321** |
| ✓(2s) | ✓ | | 81.7 | 79.5 | 0.642 |
| ✓(2s) | ✓ | ✓ | **82.7** | **81.0** | 0.720 |

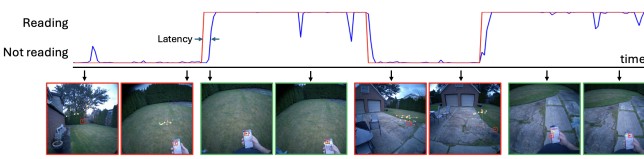

(b) Visualization using Gaze+RGB+IMU model. [Prediction/GT]

Figure 7: **Real-time detection.** We evaluate our model on alternating sequences for real-time detection. In (a), we show that (i) longer gaze sequences result in higher latency, (ii) RGB has lower latency than temporal signals (iii) adding RGB to gaze reduces the latency compared to gaze alone. We illustrate the results in (b).

sions in terms of the complementary role between gaze and RGB, but IMU does not help as much given that the dataset does not contain freeform daily activities where IMU helps the most.

Further, we also notice the differences in reading speed across different users, especially across different languages. It is possible to personalize the model by scaling the gaze to the magnitude of the reader, and empirically this solves some of the failure cases.

**Cross-language generalization.** While we only train the model on English, we find that our model generalizes well towards non-English, left-to-right texts, but less well on other languages where the reading direction is different. To circumvent this during inference, we perform 90° rotation to tackle vertical texts, and horizontally flip the gaze for right-to-left texts. We show that using gaze-only model solves the problem to a reasonable extent.

**Cross-dataset generalization.** To demonstrate the importance of collecting reading examples in freeform settings, we conduct experiments to test for generalizability across datasets. For this, we utilize EGTEA Gaze+ dataset [27], where we only use their 'reading' action labels, and treat other labels as not reading. To match the data available in EGTEA, we use 2D gaze projection at 30Hz. We conduct cross-generalization experiments where we train on one training set and evaluate on the other test set. We show that training on EGTEA with limited training samples does not generalize to in-the-wild scenarios, whereas the generalization gap for our dataset is much smaller.

### 6.4 Application: real-time reading detection

So far, we only consider atomic predictions to answer *whether* someone is reading. To extend to *when*, we simply perform predictions over time. To evaluate this task, we use the alternating sequences between reading and not reading with labeled timestamps, as shown in Figure 7. On top of the evaluation metrics, we also evaluate the latency (i.e. the duration required for a state change to be detected). Our results show that (i) there is a trade-off between gaze duration and latency; (ii) RGB has lower latency as the predictions are instantaneous, and does not rely on past detections; and (iii) combining gaze and RGB reduces the latency compared to gaze-only model.

**Localization.** To extend to *where* the user is reading, we can use the gaze point to locate the texts. As such, OCR only needs to be performed around the gaze, which results in additional compute savings. Also, the gaze scanpath can be used to estimate how much to crop the image for OCR.

**Efficient interface for OCR.** OCR comes in two phases: text detection and recognition. Using reading recognition as a low-compute interface allows OCR to run not as often, and on a smaller image each time. Furthermore, the reading detection model is designed to be small enough for on-device compute, so that images need to be transferred off-device only when reading detected, significantly reducing bandwidth requirements.

| Input | Acc | F1 | $P_{R=.9}$ |
|---|---|---|---|
| Retina images | 79.2 | 83.0 | 76.2 |
| 3D ray (d/dt) | 82.1 | 84.2 | 78.4 |
| 3D point | 80.8 | 83.3 | 77.9 |
| 3D point (d/dt) | **82.3** | **84.5** | **79.8** |
| 2D projection | 79.8 | 81.3 | 74.6 |
| Gaze + IMU | 83.9 | 85.7 | 80.0 |
| Gaze + VIO | **84.9** | **86.5** | **83.6** |

(a) Input representation

| Freq | Acc | F1 | Dur | Acc | F1 |
|---|---|---|---|---|---|
| 60 | **82.3** | **84.5** | 5 | **85.8** | **87.5** |
| 30 | 81.7 | 84.3 | 4 | 85.4 | 86.6 |
| 20 | 81.3 | 83.6 | 3 | 83.6 | 85.7 |
| 10 | 80.4 | 82.9 | 2 | 82.3 | 84.5 |
| 6 | 79.2 | 82.0 | 1 | 79.6 | 82.2 |

(b) Gaze frequency and duration

| FoV | Acc | F1 |
|---|---|---|
| 14 | **83.5** | **85.1** |
| 10 | 82.9 | 84.6 |
| 7 | 82.9 | 84.3 |
| 5 | 82.2 | 83.7 |
| 3.5 | 79.5 | 80.6 |

(c) RGB crop size

| Model | Acc | F1 |
|---|---|---|
| XS (6k) | 82.0 | 83.6 |
| S (34k) | 86.3 | 87.7 |
| M (137k) | 86.9 | 88.1 |
| L (600k) | 87.1 | 88.8 |
| XL (1M) | **88.5** | **90.1** |

(d) Model size

Table 4: **Ablation studies.** We show ablation studies for (a) the representations for gaze and IMU, (b) the gaze frequency and duration, (c) RGB crop size, and (d) model size. We fix other experiments to 60Hz, 2s, and 5° FoV using the M (137k) model, as underlined.

Figure 8: **Noise robustness.** Augmentation (red) lowers degradation.

| GT \Pred | 1 | 2 | 3 | 4 | 5 | 6 | 7 |
|---|---|---|---|---|---|---|---|
| 1 No read | 0.88 | 0.04 | 0.02 | 0.02 | 0.01 | 0.03 | 0.00 |
| 2 Walk | 0.09 | 0.85 | 0.04 | 0.01 | 0.00 | 0.00 | 0.01 |
| 3 Out loud | 0.13 | 0.02 | 0.64 | 0.17 | 0.02 | 0.01 | 0.01 |
| 4 Engaged | 0.14 | 0.02 | 0.06 | 0.54 | 0.12 | 0.01 | 0.11 |
| 5 Scan | 0.08 | 0.01 | 0.03 | 0.39 | 0.41 | 0.00 | 0.08 |
| 6 Write/type | 0.49 | 0.01 | 0.03 | 0.02 | 0.05 | 0.39 | 0.01 |
| 7 Skim | 0.13 | 0.04 | 0.05 | 0.47 | 0.15 | 0.00 | 0.16 |

Table 5: **Reading mode classification** using Gaze, RGB and IMU.

| GT \Pred | 1 | 2 | 3 | 4 | 1 | 2 | 3 | 4 |
|---|---|---|---|---|---|---|---|---|
| 1 No read | 0.77 | 0.07 | 0.04 | 0.12 | 0.83 | 0.04 | 0.03 | 0.10 |
| 2 Print | 0.07 | 0.55 | 0.29 | 0.09 | 0.08 | 0.53 | 0.25 | 0.14 |
| 3 Digital | 0.08 | 0.32 | 0.49 | 0.11 | 0.07 | 0.27 | 0.53 | 0.13 |
| 4 Objects | 0.13 | 0.28 | 0.30 | 0.29 | 0.13 | 0.18 | 0.22 | 0.47 |
| | (i) Gaze-only | | | | (ii) Gaze+IMU | | | |

Table 6: **Reading medium classification** using (i) gaze only (ii) gaze and IMU.

**Practical deployment.** We also investigate whether model of such size can be run practically. From parallel comparisons, the model can indeed comfortably run real-time on Aria Gen 2 glasses on-device, without the need to off-load the model to online computation. Given the estimated power consumption, the glasses can run for at least 4 hours continuously (inclusive of the base power consumption for basic computation, power delivery and sensor suites, and the thermal constraints). As the model runs on-device without having to send the model input and output back and forth to the server (as would have been done with, say, VLMs), the latency is negligible.

## 6.5 Ablation studies

Table 4 summarizes our results for ablation studies.

**Gaze representation.** The gaze processing pipeline involves transforming the retina images into ray angles for each eye, the intersection of which is the 3D gaze point in space, then projecting it onto the 2D image plane. We experiment using all these representations, and find that 3D gaze yields superior results, and pre-differentiating the input with respect to time leads to better generalization.

**Head pose representation.** With SLAM camaras, we can calculate the visual-intertial odometry (VIO) outputs using visual SLAM, which yields slightly better results compared to raw IMU sensors.

**Input frequency and duration.** We experiment with varying frequency and duration for eye gaze. We find that higher frequency results in better performance, but also comes with compute tradeoffs. We notice similar trends for IMU.

**RGB crop size.** While we know that human fovea only covers 2°, we find that a larger crop provides context and covers for errors in gaze estimation. However, the compute also grows quadratically.

**Model size.** We experimented other model sizes, with XS, S, M, L having 8, 16, 32, and 64 latent dimensions respectively. We also experimented with a pretrained image encoder (MobileNetV3-S) in the XL variant. We find that stronger model results in better results, and notably the S model performs surprisingly well with only 6k parameters.

**Robustness to eye tracking precision.** While our model is robust to fixed gaze offsets as we only use relative positions, noisy gaze predictions can ruin the gaze pattern. We test for the robustness to noise using our gaze-only model by adding Gaussian noise to the gaze inputs in two settings (i) only at test time and (ii) both during training (as augmentation) and testing. Our results in Figure 8 show the performance degrades with noise, and training with noise helps with robustness.

## 6.6 Extension: understanding types of reading

Many existing cognitive studies try to understand how humans read, as it is related to understanding human behavior, comprehension, and health. As mentioned, current experiments and datasets are unrepresentative of how we read. In contrast, our dataset extends to "in the wild" settings, and we

hope that our dataset will be useful in advancing the understanding of reading in the real world. Note that we use the same settings as previous experiments (2s time window), which may be limited in such fine-grained classification tasks.

**Reading mode classification.** Many studies are interested in how people read [3, 42, 28]. We conduct similar studies using our dataset. Specifically, we treat this as a 7-way classification problem (not reading, reading while walking, reading out loud, engaged reading, scanning, reading while writing/typing, skimming), and train the model for this task on our dataset. As shown in Table 5, we find that walking is an obvious category to detect (perhaps due to IMU), followed by reading out loud. Distinguishing between skimming, scanning, and engaged reading proved to be difficult.

**Reading medium classification.** Inspired by [25] that tries to answer "what" someone is reading, we also conduct similar experiments. In this case, we do not use RGB as the solution would have been trivial, and use the model to classify between four classes (not reading, print media, digital media, objects). We find that the task is difficult, and IMU helps in this case, as shown in Table 6.

## 7 Broader Impact

Always-on smart glasses raise important questions about social acceptability, both for the wearer and for the public, especially when such technologies are deployed at scale. We hope that the ideas presented in this paper can also help mitigate such concerns.

**Safety.** Sensitive personal data, such as eye gaze, introduces unique risks. Our algorithm runs fully on-device, which ensures that sensitive information does not need to leave the user's device. This is a step toward stronger privacy protections for wearers. At the same time, we acknowledge that using eye gaze as a signal creates new challenges. Eye movement can reveal intentions, interests, and even emotional states, which raises a distinct category of privacy concerns.

**Surveillance.** Our work aims to reduce reliance on invasive sensing. First, by leveraging eye gaze, we minimize the required front-camera capture to a very small patch (0.2% of the full image) rather than recording the entire scene. Second, our approach can operate solely on eye gaze data without requiring any camera input. More broadly, eye gaze offers a powerful cue about where the user is looking, which enables RGB capture to be more targeted. This reduces the risk of collecting unintended or intrusive information about bystanders. We hope that future algorithms continue in this direction.

**Data Governance.** We follow the Project Aria Research Guidelines and will release our system with a Responsible Use Policy to promote ethical research practices and to support safe deployment.

## 8 Conclusion

Motivated by use cases in contextual AI and other applications, we explore the problem of reading recognition in real-world scenarios, and present a dataset that reflects this nature. We then present a method to solve the task using three modalities, and extend the studies towards reading understanding tasks. There are vast opportunities for future work. Our dataset can be used to study the reading behavior of people in realistic settings in greater detail which links to cognitive understanding. Our proposed protocol allows for scalable future data collection using smart glasses. Additionally, model personalization to address variations in reading speed and style, along with predicting optimal modality activation for enhanced efficiency, represents another promising area for future work.

## Acknowledgements

We thank Rowan Postyeni for assistance with data collection, and Michael Goesele, Laurynas Karazija, and Lingni Ma for helpful feedback.

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
