# OpenReview forum: "Reading Recognition in the Wild"
_NeurIPS.cc/2025/Conference — NeurIPS 2025 poster_

### Official Review · Reviewer_bNbR · 2025-06-30

**Clarity:** 3
**Significance:** 2
**Originality:** 3
**Rating:** 5
**Confidence:** 3

**Summary:**

This paper proposes a method to determine whether a user's behavior is reading or not with the user's sensor information in an AR scenario. To implement the method, the paper presents a dataset containing three modalities. The paper reports that the use of the three modalities can complement each other and achieve efficient reading recognition.

**Questions:**

1.  Although the authors claim to have provided a large dataset, the total number of ids is still limited. Is it possible that some of the VLA methods are better in terms of generalization, accuracy, and stability if ft is performed on this downstream task?
2.  Notice that the encoders used in the model are small, especially the RGB encoder, is there any better performance if switching to the current popular visual encoder, such as DINO, etc.?

**Ethical Concerns:**

["Major Concern: Data privacy, copyright, and consent"]

**Final Justification:**

Thank you for the authors’ detailed response. After reading the other reviewers’ comments and the authors’ clarifications, I believe that most of my concerns have been sufficiently addressed. Additionally, I find the proposed dataset to be potentially impactful for relevant tasks. Accordingly, I have updated my score to Accept.

**Limitations:**

AR video data is highly correlated with human privacy, and it is hoped to discuss content related to user privacy.

**Quality:**

3

**Strengths And Weaknesses:**

**Strengths.**

1.  The proposed task is interesting and the video demos are generally stable.
2.  The paper presents a large-scale dataset and provides a detailed introduction, which seems to have taken a lot of effort.
3.  The paper provides detailed experimental analysis.

**Weaknesses.**

1.  Recognition result still exists lags and flickering.
2.  The methodology is a bit too simple, and the architecture is not very original or insightful.
3.  Although the paper explores different inputs and model sizes, as well as high-level comparisons with other methods, there are no experimental comparisons with other related methods (warped to the task), and it's not clear that the method has an architectural lead.

---

> ### Author Rebuttal · Authors · 2025-07-28
>
> ## Global Response:
>
> We thank all reviewers for their thoughtful evaluations and the positive reception of our work, with scores of 4 (Borderline Accept), 5 (Accept), 6 (Strong Accept), and 4 (Borderline Accept).
>
> We first highlight the key strengths identified across reviews:
>
> **Novelty and Significance of the Task:** Our paper introduces the new task of *reading recognition*. We appreciate the reviewers' acknowledgment of its impact, describing it as “interesting” (reviewer bNbR), addressing a “practical problem” (reviewer g68z), and being “timely and relevant in the context of XR and agentic AI” (reviewer YzuA). We are particularly encouraged by the recognition that this task can “enable applications like assistive tools for low-vision users” (reviewer s8g2).
>
> **Significant Dataset Contribution:** We also introduce the *Reading in the Wild* dataset. We are grateful for the reviewers’ characterization of it as “large-scale” and “detailed” (reviewer bNbR), “novel” and “comprehensive” (reviewer g68z), and a “significant contribution” (reviewer YzuA). The dataset is noted to offer a “rich and realistic benchmark for future work in egocentric activity recognition” (reviewer YzuA) and to support studies that “link to cognitive understanding” (reviewer g68z).
>
> **Method and Experimental Design:** Our multimodal reading recognition model leverages three complementary modalities, offering a practical solution to a real-world challenge. We are pleased that the reviewers found the method to be “well-motivated and clearly described” (reviewer YzuA), with the experimental design and ablations being described as “detailed,” “thorough,” and “extensive” (reviewers g68z, YzuA, bNbR).
>
> We will address specific questions and concerns in the individual responses. We are available throughout the discussion period for any additional questions or clarifications.
>
> ## Individual response [bNbR]:
>
> In this individual response, we address your specific questions and concerns in detail.
>
> **[W1] Recognition result still exists lags and flickering.**
>
> Thank you for pointing this out. We acknowledge that the model is indeed imperfect (at 86.9% accuracy) and the samples in the demo video are representative of the desired use-cases (we did not cherry-pick good results). Post-processing of output (e.g. by smoothing or filtering) may help with flickering, but we did not include it in the demo to keep it realistic. The lag (latency) and its trade-offs with performance were discussed in Figure 7. We still hope the reviewer finds it a convincing working demo for the task.
>
> **[W2] The methodology is a bit too simple, and the architecture is not very original or insightful.**
>
> We agree, though it is important to note that our primary contribution is not the model architecture itself, which builds up on standard transformer encoders, but in the identification and effective combination of different sensor modalities as inputs. The proposed architecture serves as a minimal, baseline framework for sensor fusion, with the emphasis on simplicity and ease of integration rather than architectural novelty – more complex RGB-based methods do not integrate modalities (e.g. IMU, Gaze) as easily.
>
> **[W3/Q1] Although the paper explores different inputs and model sizes, as well as high-level comparisons with other methods, there are no experimental comparisons with other related methods (warped to the task), and it's not clear that the method has an architectural lead. Is it possible that some of the VLA methods are better in terms of generalization, accuracy, and stability if ft is performed on this downstream task?**
>
> We refer the reader to Section E of the Supplementary Material for this comparison. We compared against several methods including vision-language models, action recognition models, and some alternative architectures. We note that most of these solutions (including VLAs) are not practical solutions to run on-device, which is the main aim of this paper.
>
> **[Q1 (cont’d)] Although the authors claim to have provided a large dataset, the total number of ids is still limited.**
>
> We agree that the number of participants (111) is not a large number relative to the dataset size (100 hours), unfortunately there are practical and financial constraints. To circumvent this, we focused heavily on making sure that the data we collected is generalizable. We split the train/val/test sets by participant ids (so there is no identity overlap), and have a separate Columbus eval set, to make sure that the limited ids is not a problem for practical use cases.
>
> **[Q2] Notice that the encoders used in the model are small, especially the RGB encoder, is there any better performance if switching to the current popular visual encoder, such as DINO, etc.?**
>
> We agree that the RGB encoder is particularly small, which is somewhat needed to fit on-device as it would otherwise be the computational bottleneck. We experimented using a larger RGB encoder in Table 4(d), where the XL variant uses a standard visual encoder (MobileNet) pretrained on ImageNet. This does give a performance boost, and we expect larger encoders like DINO to also provide further gains (in exchange for on-device practicality).
>
> Similarly, the L variant in the same table experimented with scaling all the encoders (both the transformer encoder and individual modality encoders), and also saw modest improvements.
>
> **[L1/Ethical Concern] AR video data is highly correlated with human privacy, and it is hoped to discuss content related to user privacy.**
>
> Thank you for pointing this out, we agree that privacy is a big concern and we have been careful about this both during data collection and how the model might be deployed.
>
> In terms of data collection, we perform controlled data collections with consenting participants instead of crowdsourcing to make sure that there are no unintentional captures of passer-bys. We also utilize EgoBlur to anonymize car plates and faces within the scene (as seen in the demo video).
>
> In terms of usage, there are several advantages of having a lightweight multimodal model on-device.
>
>
>
> 1. First, as the model runs offline and on-device, there is no need for the captured data to be transmitted off-device at all, and data can be deleted once processed.
> 2. Further, the model can work reasonably well without using the RGB modality as indicated in the main results, so the main camera does not have to be turned on in circumstances where there are privacy concerns.
> 3. More generally, eye gaze also gives a strong hint to what the user is looking at, allowing the RGB capture to focus on one place, reducing the risk of invading privacy via unintentional captures.
>
> With all being said, privacy is an important concern, especially in AR context. We hope that this paper’s ideas in using eye gaze (i) as a replacement for RGB (ii) to minimally crop RGB are positive steps in this direction. We will include this discussion in the revised version.

---

> > ### Comment · Reviewer_bNbR · 2025-08-05
> >
> > Thank you for the authors’ detailed response. After reading the other reviewers’ comments and the authors’ clarifications, I believe that most of my concerns have been sufficiently addressed. Additionally, I find the proposed dataset to be potentially impactful for relevant tasks. Accordingly, I have updated my score to **Weak Accept**.

---

> > > ### Author Response · Authors · 2025-08-08
> > >
> > > We thank the reviewer for the reply. We are glad to have addressed the concerns, and pleased to know that the reviewer also thinks the dataset can be "potentially impactful for relevant tasks", and is willing to update the score.
> > >
> > > From our end, the score does not appear to have been finalized. Could you help check if the "Rating" has been updated accordingly? Many thanks.

---

### Official Review · Reviewer_YzuA · 2025-07-04

**Clarity:** 4
**Significance:** 4
**Originality:** 4
**Rating:** 6
**Confidence:** 5

**Summary:**

This paper presents a novel method for detecting reading behavior in smartglass wearers using multimodal data. The authors introduce a dataset comprising 100 hours of egocentric video, eye gaze, and IMU data in diverse scenarios. They propose a transformer-based model capable of leveraging each modality individually or in combination. The work is timely, well-executed, and offers strong empirical results.

**Questions:**

Can the model be deployed on the glasses? What are the implications for battery life and latency?

**Ethical Concerns:**

["NO or VERY MINOR ethics concerns only"]

**Final Justification:**

My question on computational efficiency has been answered, and I am satisfied with it. Recommend acceptance.

**Limitations:**

Discussion on limitations is sufficient

**Paper Formatting Concerns:**

None noted.

**Quality:**

4

**Strengths And Weaknesses:**

Strengths:

+ Timely and relevant task in the context of XR and agentic AI, where understanding user behavior such as reading can enable more intelligent and context-aware systems.

+ The proposed dataset is a significant contribution, offering a rich and realistic benchmark for future work in egocentric activity recognition.

+ The transformer-based approach is well-motivated and clearly described. The ability to handle multiple modalities flexibly is a strong point.

+ Strong performance, achieving over 90% precision and 85% recall, indicating practical viability.

+ Extensive experiments, ablation studies, and thoughtful discussion on generalization to related tasks beyond reading detection.

+ Promised release of code, data, and models.


Minor Weaknesses:

+ The paper lacks discussion on computational efficiency and real-time deployment feasibility on the glasses. For instance, can the model run on the Aria Glasses? What are the implications for battery life and latency?

---

> ### Author Rebuttal · Authors · 2025-07-28
>
> ## Global Response:
>
> We thank all reviewers for their thoughtful evaluations and the positive reception of our work, with scores of 4 (Borderline Accept), 5 (Accept), 6 (Strong Accept), and 4 (Borderline Accept).
>
> We first highlight the key strengths identified across reviews:
>
> **Novelty and Significance of the Task:** Our paper introduces the new task of *reading recognition*. We appreciate the reviewers' acknowledgment of its impact, describing it as “interesting” (reviewer bNbR), addressing a “practical problem” (reviewer g68z), and being “timely and relevant in the context of XR and agentic AI” (reviewer YzuA). We are particularly encouraged by the recognition that this task can “enable applications like assistive tools for low-vision users” (reviewer s8g2).
>
> **Significant Dataset Contribution:** We also introduce the *Reading in the Wild* dataset. We are grateful for the reviewers’ characterization of it as “large-scale” and “detailed” (reviewer bNbR), “novel” and “comprehensive” (reviewer g68z), and a “significant contribution” (reviewer YzuA). The dataset is noted to offer a “rich and realistic benchmark for future work in egocentric activity recognition” (reviewer YzuA) and to support studies that “link to cognitive understanding” (reviewer g68z).
>
> **Method and Experimental Design:** Our multimodal reading recognition model leverages three complementary modalities, offering a practical solution to a real-world challenge. We are pleased that the reviewers found the method to be “well-motivated and clearly described” (reviewer YzuA), with the experimental design and ablations being described as “detailed,” “thorough,” and “extensive” (reviewers g68z, YzuA, bNbR).
>
> We will address specific questions and concerns in the individual responses. We are available throughout the discussion period for any additional questions or clarifications.
>
> ## Individual response [YzuA]:
>
> In this individual response, we address your specific questions and concerns in detail.
>
> **[W1] The paper lacks discussion on computational efficiency and real-time deployment feasibility on the glasses. For instance, can the model run on the Aria Glasses? What are the implications for battery life and latency?**
>
> Thank you for the comment and questions, we agree that this is an important aspect. We can confirm that the algorithm is intentionally designed to be lightweight enough to run on-device.
>
> We have run similar models in terms of network capacity and architecture on the Aria Gen 2 glasses, and got the following results:
>
> * Computing power: the model can indeed comfortably run on Aria Gen 2 glasses on-device, without the need to off-load the model to online computation
> * Battery: given the estimated power consumption, the glasses can run for at least 4 hours continuously (inclusive of the base power consumption for basic computation, power delivery and sensor suites, and the thermal constraints).
> * Communication: as the model runs on-device without having to send the model input and output back and forth to the server (as would have been done with, say, VLMs), the latency is negligible
> * Runtime: we are unable to estimate the model inference runtime on-device as this requires integrating with on-device ML accelerators, but from parallel comparisons with similar deployments, models of this size (130k params) can easily run on-device in real-time.
>
> We will include this discussion in the revised version.

---

### Official Review · Reviewer_g68z · 2025-07-05

**Clarity:** 3
**Significance:** 3
**Originality:** 4
**Rating:** 5
**Confidence:** 3

**Summary:**

The paper addresses a practical problem for always-on smart glasses: efficiently detecting when users are reading to enable contextual AI. The authors introduce a 100-hour multimodal dataset collected using Project Aria glasses and propose a transformer-based model that combines eye gaze, RGB crops, and IMU data to achieve 86.9% accuracy on reading detection.  The proposed dataset can be used to study the reading behavior of people in realistic settings in greater detail which links to cognitive understanding.

**Questions:**

Please check the weakness part.

**Ethical Concerns:**

["NO or VERY MINOR ethics concerns only"]

**Quality:**

3

**Strengths And Weaknesses:**

[Strength]

- This paper introduced a novel problem and datasets to the communities, which has good positioning within the broader context of contextual AI

- The introduced dataset is comprehensive, with 100 hours of diverse data across 111 participants, and diverse reading modes and environments.

- The authors has thorough experimental design and comprehensive analysis of model components, input representations, and hyperparameters.

[Weakness]

- There might be language bias: Training primarily on English, though cross-language experiments partially address this

- There might be reading material bias: May not cover all types of real-world reading scenarios

---

> ### Author Rebuttal · Authors · 2025-07-28
>
> ## Global Response:
>
> We thank all reviewers for their thoughtful evaluations and the positive reception of our work, with scores of 4 (Borderline Accept), 5 (Accept), 6 (Strong Accept), and 4 (Borderline Accept).
>
> We first highlight the key strengths identified across reviews:
>
> **Novelty and Significance of the Task:** Our paper introduces the new task of *reading recognition*. We appreciate the reviewers' acknowledgment of its impact, describing it as “interesting” (reviewer bNbR), addressing a “practical problem” (reviewer g68z), and being “timely and relevant in the context of XR and agentic AI” (reviewer YzuA). We are particularly encouraged by the recognition that this task can “enable applications like assistive tools for low-vision users” (reviewer s8g2).
>
> **Significant Dataset Contribution:** We also introduce the *Reading in the Wild* dataset. We are grateful for the reviewers’ characterization of it as “large-scale” and “detailed” (reviewer bNbR), “novel” and “comprehensive” (reviewer g68z), and a “significant contribution” (reviewer YzuA). The dataset is noted to offer a “rich and realistic benchmark for future work in egocentric activity recognition” (reviewer YzuA) and to support studies that “link to cognitive understanding” (reviewer g68z).
>
> **Method and Experimental Design:** Our multimodal reading recognition model leverages three complementary modalities, offering a practical solution to a real-world challenge. We are pleased that the reviewers found the method to be “well-motivated and clearly described” (reviewer YzuA), with the experimental design and ablations being described as “detailed,” “thorough,” and “extensive” (reviewers g68z, YzuA, bNbR).
>
> We will address specific questions and concerns in the individual responses. We are available throughout the discussion period for any additional questions or clarifications.
>
> ## Individual response [g68z]:
>
> In this individual response, we address your specific questions and concerns in detail.
>
> **[W1] There might be language bias: Training primarily on English, though cross-language experiments partially address this**
>
> Thank you for pointing these out. We acknowledge the fundamental limitation of our dataset that the training set only involves English, posing a bias in the dataset that limits the usefulness towards different languages.
>
> While we are unable to expand this to all languages due to resource constraints, we have been mindful about the generalization to different languages including (i) the manual augmentation method to partially circumvent this limitation and (ii) a benchmark for testing generalization to other languages.
>
> We hope that this is a good step in this direction, and to encourage further work both to diversify the dataset and find better methods to improve language generalization.
>
> **[W2] There might be reading material bias: May not cover all types of real-world reading scenarios**
>
> We agree, and while we try our best to diversify the reading materials by having a comprehensive data collection protocol as indicated in the Supplementary Material, they are likely mostly concentrated in the western world. We tried to circumvent this by collecting Seattle and Columbus subsets completely separately, but we acknowledge the limitations given that both subsets are still collected in the USA.
>
> That said, we do want to highlight our efforts in collecting diverse reading materials, as well as diversity in the dataset as a whole in Sections A-C of the Supplementary.

---

### Official Review · Reviewer_s8g2 · 2025-07-06

**Clarity:** 3
**Significance:** 3
**Originality:** 3
**Rating:** 4
**Confidence:** 3

**Summary:**

The authors introduce reading recognition—a task to detect when a user is reading—for contextual AI in smart glasses. They present the Reading in the Wild dataset (100 hours of egocentric video, gaze, and IMU data across diverse real-world scenarios) and a lightweight transformer model leveraging RGB gaze crops, eye gaze, and head pose (IMU). The modalities are shown to be complementary, with combined use achieving 86.9% accuracy. The dataset and model enable applications like assistive tools for low-vision users and extend to classifying reading modes (e.g., skimming) and mediums (e.g., print).

**Questions:**

Could personalizing models to individual reading styles (e.g., speed, gaze patterns) improve edge-case performance?  And for right-to-left languages, would training with minimal flipped-gaze samples (vs. inference-time flipping) enhance generalization?

**Ethical Concerns:**

["NO or VERY MINOR ethics concerns only"]

**Limitations:**

Training data limited to English; generalization to non-LTR scripts requires manual augmentation. And fixed 2s input may not capture long reading sessions or rapid transitions.

**Quality:**

3

**Strengths And Weaknesses:**

The authors introduce reading recognition—a task to detect when a user is reading—for contextual AI in smart glasses. They present the Reading in the Wild dataset (100 hours of egocentric video, gaze, and IMU data across diverse real-world scenarios) and a lightweight transformer model leveraging RGB gaze crops, eye gaze, and head pose (IMU). The modalities are shown to be complementary, with combined use achieving 86.9% accuracy. The dataset and model enable applications like assistive tools for low-vision users and extend to classifying reading modes (e.g., skimming) and mediums (e.g., print).

Model struggles with vertical (Chinese: 35.5% accuracy) and right-to-left (Arabic: 21.0%) texts without augmentations (Table 3b). Reading mode (e.g., skimming vs. scanning) and medium (object vs. digital) tasks underperform (Tables 5–6), suggesting the 2s window is insufficient.

---

> ### Author Rebuttal · Authors · 2025-07-28
>
> ## Global Response:
>
> We thank all reviewers for their thoughtful evaluations and the positive reception of our work, with scores of 4 (Borderline Accept), 5 (Accept), 6 (Strong Accept), and 4 (Borderline Accept).
>
> We first highlight the key strengths identified across reviews:
>
> **Novelty and Significance of the Task:** Our paper introduces the new task of *reading recognition*. We appreciate the reviewers' acknowledgment of its impact, describing it as “interesting” (reviewer bNbR), addressing a “practical problem” (reviewer g68z), and being “timely and relevant in the context of XR and agentic AI” (reviewer YzuA). We are particularly encouraged by the recognition that this task can “enable applications like assistive tools for low-vision users” (reviewer s8g2).
>
> **Significant Dataset Contribution:** We also introduce the *Reading in the Wild* dataset. We are grateful for the reviewers’ characterization of it as “large-scale” and “detailed” (reviewer bNbR), “novel” and “comprehensive” (reviewer g68z), and a “significant contribution” (reviewer YzuA). The dataset is noted to offer a “rich and realistic benchmark for future work in egocentric activity recognition” (reviewer YzuA) and to support studies that “link to cognitive understanding” (reviewer g68z).
>
> **Method and Experimental Design:** Our multimodal reading recognition model leverages three complementary modalities, offering a practical solution to a real-world challenge. We are pleased that the reviewers found the method to be “well-motivated and clearly described” (reviewer YzuA), with the experimental design and ablations being described as “detailed,” “thorough,” and “extensive” (reviewers g68z, YzuA, bNbR).
>
> We will address specific questions and concerns in the individual responses. We are available throughout the discussion period for any additional questions or clarifications.
>
> ## Individual response [s8g2]:
>
> In this individual response, we address your specific questions and concerns in detail.
>
> **[W1/L1] Model struggles with vertical (Chinese: 35.5% accuracy) and right-to-left (Arabic: 21.0%) texts without augmentations (Table 3b). Training data limited to English; generalization to non-LTR scripts requires manual augmentation.**
>
> Thank you for pointing these out. We acknowledge the fundamental limitation of our dataset that the training set only involves English, posing a bias in the dataset that limits the usefulness towards different languages.
>
> While we are unable to expand this to all languages due to resource constraints, we have been mindful about the generalization to different languages including (i) the manual augmentation method to partially circumvent this limitation and (ii) a benchmark for testing generalization to other languages.
>
> We hope that this is a good step in this direction, and to encourage further work both to diversify the dataset and find better methods to improve language generalization.
>
> **[L2] Fixed 2s input may not capture long reading sessions or rapid transitions.**
>
> Yes, the observation is correct, though the answer is a bit more nuanced. We investigated the input duration and presented the results in Table in Figure 7(a) in the Main Paper. In summary, there is a fundamental trade-off between being able to capture long reading sessions (better with longer input) and being able to capture rapid transitions (better with shorter input).
>
> While there is no perfect answer for a single model because of this trade-off, we chose 2s as a practical middle ground between the two. With reference to human vision, we also think this choice of 2s time window is reasonable given studies like [1, 2, 3], where an average fixation during reading is 200-250ms and saccade 20-40ms. We think it is possible to create an ensemble of models with different input durations.
>
> [1] Sereno and Rayner. Measuring word recognition in reading: eye movements and event-related potentials. Trends Cogn Sci. 2003.
>
> [2] Frey et al. An Eye Fixation-Related Potential Study in Two Reading Tasks: Reading to Memorize and Reading to Make a Decision. Brain Topography, 2018.
>
> [3] Biedert et al. A robust realtime reading-skimming classifier. In ETRA, 2012.
>
>
>
> **[W2] Reading mode (e.g., skimming vs. scanning) and medium (object vs. digital) tasks underperform (Tables 5–6), suggesting the 2s window is insufficient.**
>
> This answer follows similarly to the previous answer in [L2]. For the reading mode and medium classification, we agree that using a longer time window is likely to improve the model. We did some experiments specifically for the reading mode and medium classification tasks. We observed marginal increase in accuracy at the cost of extra compute and latency when increasing the window to 3-5s.
>
> **[Q1] Could personalizing models to individual reading styles (e.g., speed, gaze patterns) improve edge-case performance?**
>
> Thank you for your insight, this is an interesting question. We agree personalization is a good way to improve performance.
>
> To experimentally verify this, we identified a test case that failed because the participant’s reading speed was too slow. We then experimented scaling the input velocity of this specific reader so that the average velocity is the same as what is obtained in the entire dataset, i.e. new_velocity = old_velocity * (dataset_average / user_average).
>
> The scaling factor is calculated to be (0.6173 m/s / 0.3951 m/s) = 1.562, and we use the scaled velocity as input without modifying the model. As a result, we found that the F1-score of this particular script increased from 35.6 to 61.4, verifying the effectiveness of personalization.
>
> We will include this experiment as part of the main paper, we again appreciate the comment.
>
> **[Q2] And for right-to-left languages, would training with minimal flipped-gaze samples (vs. inference-time flipping) enhance generalization?**
>
> Thank you for the question. We did several experiments with data augmentation, both on rotation to handle vertical languages, and flipping to handle right-to-left languages.
>
> We find that while both train-time augmentations improve the results on the edge cases. However, the rotation augmentation has minimal effect on the overall model performance while the flipping augmentation has a more severe impact by enabling false positives on non-reading activities (such as shaking heads), and also a lowered recall on normal reading scenarios.
>
> The model we reported actually did include rotation augmentation as part of its training in 10% of its samples. We think the left-to-right direction bias is actually beneficial to the model, and there is a trade-off between main model performance and generalization to less common reading styles.
>
> We realized we did not include this part in the paper, so we will make this clear in the final edition of this paper, as well as in the code release.

---

### Author Response · Authors · 2025-08-07

We thank everyone for their responses so far. We hope to have addressed all reviewers' concerns. If you still have any questions or concerns, we are still available to respond, but please note that we only have until Aug 8, 11.59pm AoE to discuss.

If there are no further concerns, we would appreciate updating the post-rebuttal justification and acknowledgement accordingly.

---

### Decision · Program_Chairs · 2025-09-17

**Decision:**

Accept (poster)

**Comment:**

The paper introduces a new task of reading recognition in the wild, supported by a 100h multimodal dataset and a transformer-based model combining RGB gaze crops, eye gaze, and head pose. Reviewers found the task novel, the dataset significant, and the experiments thorough. Main concerns were dataset bias (English-centric, weak on non-left-to-right scripts), limited generalization, and flicker/lag in demos. Ethical reviews echoed risks around inclusivity and privacy but noted proper consent and plans for responsible release. Overall, the strengths of the dataset and task highlight the importance of the work, leading to acceptance.